# Papillary Squamotransitional Cell Carcinoma of the Uterine Cervix with Atypical Presentation: A Case Report with a Literature Review

**DOI:** 10.3390/medicina58121838

**Published:** 2022-12-14

**Authors:** Angel Yordanov, Milen Karaivanov, Stoyan Kostov, Yavor Kornovski, Yonka Ivanova, Stanislav Slavchev, Venelina Todorova, Mariela Vasileva-Slaveva

**Affiliations:** 1Department of Gynaecological Oncology, Medical University Pleven, 5800 Pleven, Bulgaria; 2Department of General and Clinical Pathology, University Hospital “Dr. Georgi Stranski”, 5800 Pleven, Bulgaria; 3Department of Gynecology, St. Anna University Hospital, Medical University—Varna “Prof. Dr. Paraskev Stoyanov”, 9000 Varna, Bulgaria; 4Imaging Department, University Hospital “Dr. Georgi Stranski”, 5800 Pleven, Bulgaria; 5Department of Breast Surgery, Shterev Hospital, 1000 Sofia, Bulgaria; 6Research Institute, Medical University Pleven, 5800 Pleven, Bulgaria

**Keywords:** squamous cell carcinoma, papillary squamotransitional cell carcinoma, lymph node metastasis, diagnosis, treatment

## Abstract

*Introduction*: Cervical cancer is the fourth most prevalent malignancy and the fourth leading cause of cancer-related death in women around the world. Histologically, squamous cell carcinoma (SCC) is the most common form of cervical cancer. SCC has several subtypes, and one of the rarest is papillary squamotransitional cell carcinoma (PSCC). In general, PSCC is believed to have a similar course and prognosis to typical SCC, with a high risk of late metastasis and recurrence. *Case report*: We discuss the case of a 45-year-old patient diagnosed with PSCC who was admitted to our department in December 2021. The clinical manifestations were pelvic discomfort and lymphadenopathy throughout the body. On admission, all laboratory values, with the exception of C-Reactive Protein (CRP) at 22.35 mg/L and hemoglobin (HGB) at 87.0 g/L, were normal. The clinical and ultrasound examination revealed a painful formation with indistinct borders in the right portion of the small pelvis. Following dilation and curettage, a Tru-Cut biopsy of the inguinal lymph nodes was performed. The investigation histologically indicated PSCC. MRI of the small pelvis showed an endophytic tumor in the cervix with dimensions of 35/26 mm and provided data for bilateral parametrial infiltration; a hetero-intensive tumor originating from the right ovary and involving small intestinal loops measuring 90/58 mm; and generalized lymphadenopathy and peritoneal metastases in the pouch of Douglass. The FIGO classification for the tumor was IVB. The patient was subsequently referred for chemotherapy by the tumor board’s decision. *Discussion*: Despite the generally good prognosis of SCC, PSCC is a rare and aggressive subtype. It is usually diagnosed at an advanced stage and has a poor prognosis. *Conclusions*: PSCC is a rare subtype of SCC, and its diagnosis and treatment are challenging.

## 1. Introduction

Gynecologic malignancies represent 16.5% of all 8.2 million reported new cancer cases in female [1]. Despite therapeutic advances in the last few decades, the clinical outcomes of gynecological cancers are still poor, presenting high incidence and cancer-related mortality [2].

Cervical cancer is the fourth most prevalent malignancy and the fourth leading cause of cancer-related death in women globally [3]. It is also one of the three most common cancers that affect women younger than 45 years old [4]. Recently, the incidence of cervical cancer has been reduced, especially in developed countries, because of the widespread use of primary and secondary prevention strategies [5]. In developing countries, cervical cancer is still a major health problem and is usually caused by a persistent infection from high-risk human papillomavirus (HPV) [6]. Squamous cell carcinoma (SCC) is the most prevalent histological variety (75%), followed by adenocarcinoma (10–25%) and all other rare variants including adenosquamous carcinoma and neuroendocrine carcinoma [7]. Several subtypes of SCC occur with varied frequency: keratinizing, non-keratinizing, basaloid, verrucous, warty, papillary, lymphoepithelioma-like, and squamotransitional [8]. The incidence of papillary squamotransitional cell carcinoma (PSCC) is estimated to represent 1.6% of all cervical malignancies [9]. Due to its limited incidence, doctors are unfamiliar with this tumor, and little is known about its clinical behavior and progression. It is believed that late metastasis and late local recurrences are possible [10]. 

We present a case of aggressive PSCC in a 45-year-old patient.

## 2. Case Report 

A 45-year-old female presented to our clinic in December 2021 with pelvic pain and a developed echo-heterogeneous formation measuring 80/70 mm and emanating from the right ovary. A thorough medical history revealed iliofemoral thrombophlebitis of the right calf eight months prior to admission; arterial hypertension on systemic therapy (seemingly well-controlled); and no menstrual abnormalities to date. The clinical examination indicated a painful development with indistinct borders and dimensions of 7/7 cm in the right portion of the small pelvis. The ultrasound scan revealed that the right ovary contained a 90/50 mm echo-heterogeneous tumor. Aside from the levels of C-Reactive Protein (CRP) and hemoglobin (HGB) of 22.35 and 87, respectively, the patient’s laboratory results were normal. A primary tumor of the right ovary was suspected. 

Upon consultation with a vascular surgeon, the iliofemoral thrombophlebitis had been completely resolved. MRI of the small pelvis revealed an endophytic tumor in the cervix with dimensions of 35/26 mm and evidence of bilateral, initial parametrial infiltration (Figure 1A); a hyperintense tumor originating from the right ovary and involving small intestinal loops measuring 90/58 mm (Figure 1B); enlarged pelvic lymph nodes (21.9/15.1 mm) on the right and (30.8/19.5 mm) on the left (Figure 1C); and multiple, enlarged inguinal lymph nodes up to 29 mm in size on the right and up to 40 mm in size on the left (Figure 1D). 

Dilation and curettage were performed, which revealed the following findings: endocervical tissue with groups of glands exhibiting moderate structural and cellular atypia (GIN—low grade) and diffuse infiltration of tumor cells consisting of micropapillae, nests, and strands, while the cells had eosinophilic and light cytoplasm, large hyperchromatic nuclei with euchromatin, and oval and polygonal shapes. Figure 2 is an illustration of the pathogenic findings (Figure 2A).

The following immunohistochemical markers were examined:

p16 (+) in over 90% of the tumor cells (Figure 2B); 

p63 (+) in over 80% of the cells (Figure 2C); 

CK7 (+) in over 90% of the cells (Figure 2D); 

CK20 (+) in over 50% of the cells (Figure 2E); 

CEA (+) in the peripheral layers of the tumor nests (Figure 2F); 

Vimentin (−); 

GATA3 (−); 

ER (−); 

PAS/Alc. blue (−).

The tumor was diagnosed as PSCC.

CT scans of the chest, abdomen, and pelvis were performed due to the patient’s enlarged pelvic and inguinal lymph nodes. The scans showed generalized lymphadenopathy in the mediastinal, paraaortic, and pelvic lymph nodes and a formation in right ovary (Figure 3).

A laparoscopic biopsy of the right ovary was offered to the patient, but she refused it. A Tru-Cut biopsy of the patient’s inguinal lymph nodes was performed; consequently, adipose and fibrous tissue with carcinoma infiltration and a micropapillary growth pattern composed of tumor cells with moderate nuclear polymorphism (Figure 4A), as well as areas with necrosis, were acquired.

The immunohistochemical study of other markers revealed WT1-negativity in the tumor cells, with positive internal control in the stroma; p63 was positive in 80–90% of the tumor cells (Figure 4B); and p53 was uninterpretable.

Despite the patient’s refusal of ovarian biopsy, the tumor board staged the tumor as IVB by FIGO. The patient was referred for additional chemotherapy treatment. After administering four cycles of first-line chemotherapy with avastin, carboplatin, and paclitaxel over a period of 21 days, a restaging CT of the entire body revealed a partial response (PR) (Figure 5 and Figure 6). 

A decision was made to maintain the patient on the same chemotherapy regimen due to the presence of a partial response.

## 3. Discussion

Transitional cell carcinomas were first reported in the ovaries and the fallopian tubes and later in the endometrium and cervix [11,12]. Marsh was the first to describe it in 1952. In 1986, Randall et al. proposed papillary squamous cell carcinomas as a separate subtype [9,13], which was acknowledged in 2003 by the WHO cervical cancer histological classification [14].

The incidence of PSCC is estimated to be 1.6% but given the number of unpublished and unrecognized cases it may be higher [15].

Koenig et al. divide PSCC into three types depending on the papillary component, each with a different frequency [16]:Predominantly squamous (28.1%);Mixed squamous and transitional (50%);Predominantly transitional (21.9%).

All other papillary lesions of the cervix (Table 1) [17] should be excluded when PSCC is diagnosed.

Determining stromal invasion after biopsy is sometimes very difficult because of the complex surface papillary architecture of these lesions [18], and this can lead to misdiagnosis and inadequate follow-up treatment for the patient.

In terms of risk factors for onset and clinical behavior, PSCC is considered to be similar to classical SCC, with the potential for late local recurrences and metastasis [10,15]. There is currently no specific treatment for PSCC [15], and it should be treated as SCC according to the stage of the disease [9,16].

There are several cases of very early recurrence described in the literature, with some presenting up to several months after surgery [15,19], which indicates its aggressive course. In addition, there are cases of an advanced stage at the time of the diagnosis [17,20,21,22,23,24].

To the best of our knowledge, there is only one case described in the literature to date in which PSCC has manifested ovarian metastasis [17]. 

In our case, the initial manifestations of the disease are pelvic pain caused by ovarian development and iliofemoral thrombophlebitis of the right calf, which we assume to be the result of the oncological process. The absence of aberrant uterine bleeding and the normal macroscopic appearance of the cervix led to a misdiagnosis of a tubo-ovarian abscess. The diagnosis was initially determined by pelvic MRI and confirmed by pathological analysis. Immunohistochemical testing validated the diagnosis of primarily transitional PSCC. To search for distant metastases, a CT scan of the entire body was performed; consequently, metastases were discovered in the mediastinal and paraaortic lymph nodes. None were discovered in the organ parenchyma. A biopsy of the inguinal lymph nodes was conducted to confirm the diagnosis (due to the lack of histology of the right ovary, we could not completely rule out ovarian malignancy, but the CT results after four courses of chemotherapy confirmed our initial opinion). After analyzing the results of the inguinal lymph nodes, we determined that the cervical tumor had spread to the inguinal lymph nodes. 

## 4. Conclusions

PSCC is a rare tumor with unknown prevalence and clinical behavior for which adequate treatment is unavailable. The small number of reported cases makes it impossible to establish strategies for recognizing and treating this malignancy. For this reason, every new case must be recorded so that we can fill the gaps in our knowledge.

## Figures and Tables

**Figure 1 medicina-58-01838-f001:**
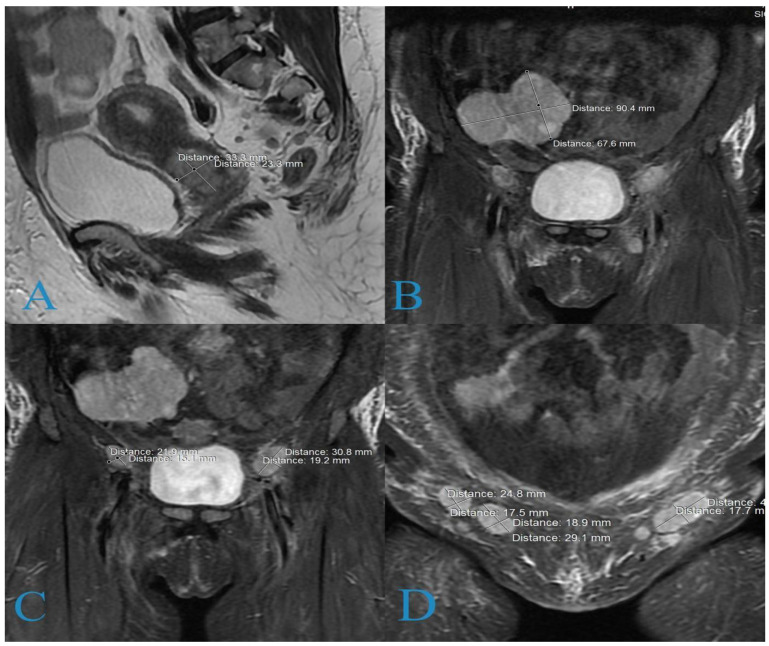
Magnetic resonance imaging: (**A**) cervical formation; (**B**) ovarian formation; (**C**) pathologically enlarged pelvic lymph nodes; (**D**) bilaterally pathologically enlarged inguinal lymph nodes forming a cluster on the left side.

**Figure 2 medicina-58-01838-f002:**
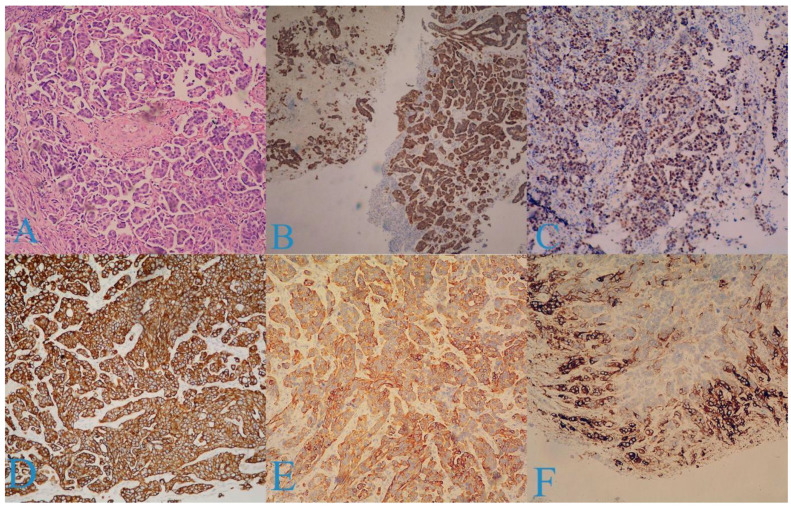
Histologic findings: (**A**) tumor consisting of micropapillae, nests, and strands (×25 H&E); (**B**) p16 (+) in over 90% of the tumor cells; (**C**) p63 (+) in over 80% of the cells; (**D**) CK7 (+) in over 90% of the cells; (**E**) CK20 (+) in over 50% of the cells; (**F**) CEA (+) in the peripheral layers of tumor nests.

**Figure 3 medicina-58-01838-f003:**
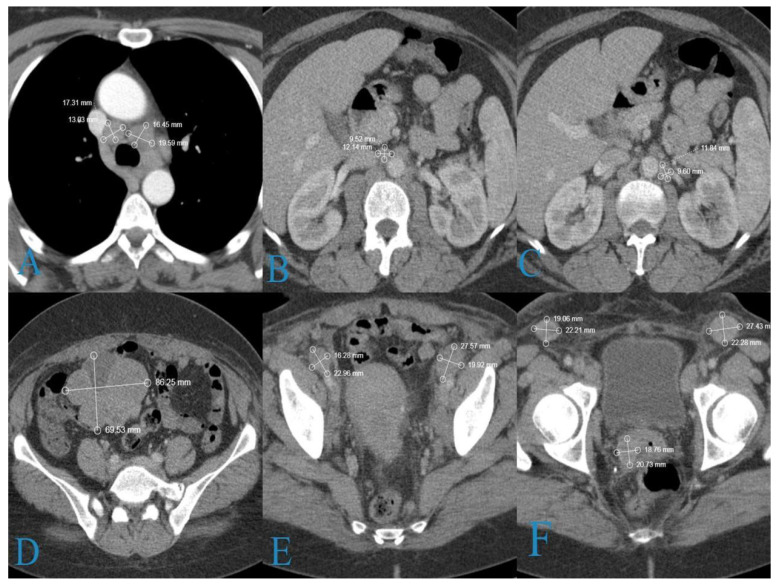
CT findings: (**A**) low paratracheal group (19/14 mm); (**B**) right paraaortic lymph nodes (11/9 mm); (**C**) left paraaortic lymph nodes (20/10 mm); (**D**) right ovarian tumor formation (87/69 mm); (**E**) common iliac lymph nodes (23/17 mm on the right and 28/18 mm on the left); (**F**) soft tissue lesion in the cervix (35/25) mm and inguinal lymph nodes (22/18 mm on the right and 27/23 mm on the left).

**Figure 4 medicina-58-01838-f004:**
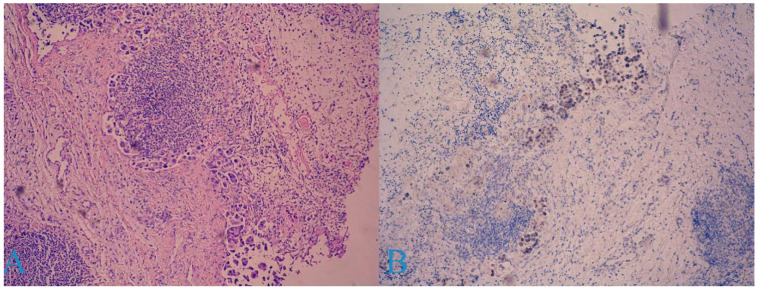
Histological findings of a metastatic inguinal lymph node: (**A**) carcinoma tissue with micropapillary growth pattern; (**B**) p63—positive in 80–90% of tumor the cells.

**Figure 5 medicina-58-01838-f005:**
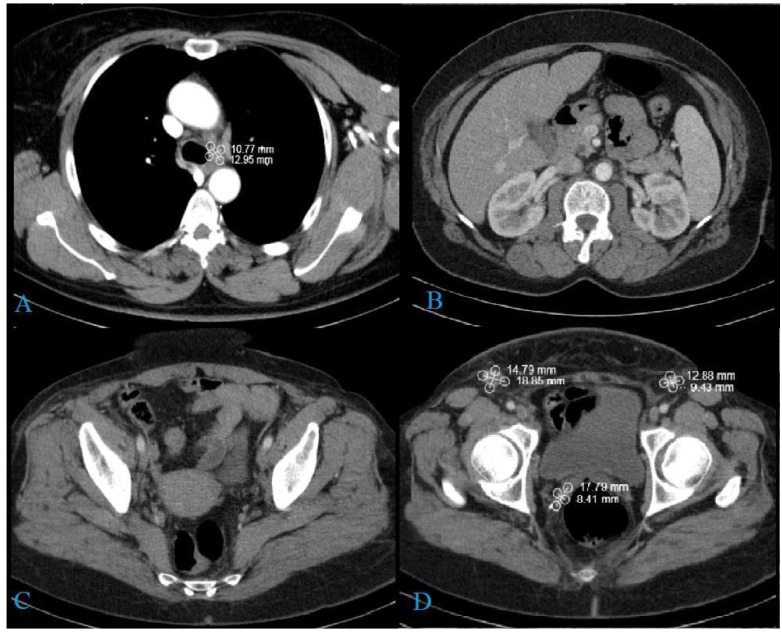
Restaged CT findings: (**A**) 30% decrease in the pathologically enlarged paratracheal lymph nodes; (**B**) the para aortic lymph nodes are not visible (all with normal size); (**C**) the common iliac lymph nodes are also not visible; (**D**) soft tissue lesion in the cervix and the inguinal lymph nodes again present size reduction of over 30%.

**Figure 6 medicina-58-01838-f006:**
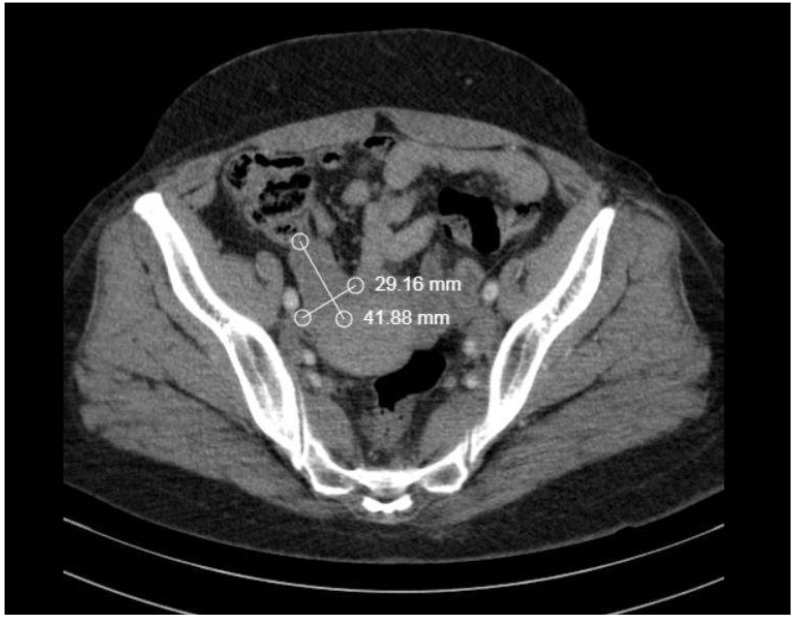
The restaging CT of the right ovary: sizes were reduced more than 50%.

**Table 1 medicina-58-01838-t001:** Differential diagnosis of papillary lesions of the cervix [17].

Benign	Malignant
Condyloma acuminatum	Warty squamous cell carcinoma
Squamous papilloma	Verrucous carcinoma
Cervical intraepithelial neoplasia with papillary configuration	Transitional cell carcinoma of the endometrium and endometrial adenocarcinoma
Papillary adenofibroma	Villoglandular papillary adenocarcinoma

## Data Availability

The authors declare that all related data concerning this research and its researchers can be made available by contacting the corresponding author via email.

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
