# Peer review of "Papillary Squamotransitional Cell Carcinoma of the Uterine Cervix with Atypical Presentation: A Case Report with a Literature Review"

_medicina, 2022, doi:10.3390/medicina58121838_

Round 1

Reviewer 1 Report

I read with great interest the Manuscript titled "Papillary Squamotransitional Cell Carcinoma of the Uterine Cervix with Atypical Presentation. A Case Report with a Literature Reviewwhich falls within the aim of the Journal.

In my honest opinion, the topic is interesting enough to attract the readers’ attention. Nevertheless, authors should clarify some point and improve the discussion citing relevant and novel key articles about the topic.

-First typos errors must be corrected.

-Moreover, the whole text should be corrected by a native English speaker in order to make the work clearer and more readable.

-The introduction should be extended and completed. I find interesting a reference to the efforts made for the prevention and early diagnosis of gynecological cancers (see PMID: 36141217).

- Authors should revise the text and correct the acronyms contained, paying attention to report in full their meaning the first time the acronym is used in the text. (i.e., line 26).

- I suggest also authors to not repeat some concepts more than once and to organize better the introduction. (Discussion and conclusions are overlapping).

-Discussions can be expanded and improved by citing relevant articles (I suggest authors to read and insert in references the following article PMID:  35742340; 35455328).

-What are the strengths and limitations of this manuscript?

Considered all this points, in my opinion, it deserves the priority to be published after minor revisions.

Author Response

I read with great interest the Manuscript titled "Papillary Squamotransitional Cell Carcinoma of the Uterine Cervix with Atypical Presentation. A Case Report with a Literature Review" which falls within the aim of the Journal.

In my honest opinion, the topic is interesting enough to attract the readers’ attention. Nevertheless, authors should clarify some point and improve the discussion citing relevant and novel key articles about the topic.

Authors’reply: Thank you very much for reviewing this article. I greatly appreciate you taking the time to review it!

-First typos errors must be corrected.

Authors’reply: Done as recommended

-Moreover, the whole text should be corrected by a native English speaker in order to make the work clearer and more readable.

Authors’reply: Done as recommended. Native English speaker rewrote the whole manuscript and that’s why we did not use a track changes.

-The introduction should be extended and completed. I find interesting a reference to the efforts made for the prevention and early diagnosis of gynecological cancers (see PMID: 36141217).

Authors’reply: Done as recommended. We incorporated the following text:

 The gynecologic malignancies represent 16.5% of all 8.2 million reported new cancer cases in female (1). Despite therapeutic advances in the last few decades, the clinical outcome of the gynecological cancers is still poor and they are with high incidence and cancer-related mortality (2).

- Authors should revise the text and correct the acronyms contained, paying attention to report in full their meaning the first time the acronym is used in the text. (i.e., line 26).

Authors’reply: Done as recommended

- I suggest also authors to not repeat some concepts more than once and to organize better the introduction. (Discussion and conclusions are overlapping).

Authors’reply: Done as recommended. We changed the conclusin section to:

PSCC is a rare tumor with unknown prevalence, clinical behavior, and treatment. The small number of reported cases, makes it impossible to establish strategies for recognizing and treating this malignancy For this reason, every new case must be recorded so that we can fill up the gaps in our knowledge

-Discussions can be expanded and improved by citing relevant articles (I suggest authors to read and insert in references the following article PMID:  35742340; 35455328).

Authors’reply: We used these two articles in introduction section

-What are the strengths and limitations of this manuscript?

Authors’reply: It is very difficult to mark the strengths and limitations of case report. In our oppinion the strengths are the images we have and the limitation is the short follow-up

Reviewer 2 Report

This is a very interesting case report describing a rare subtype of cervical cancer, that it is important to know.

The story of patients, diagnosis and treatment are weel described except for radiotherapy: in the abstract it is reported that patient underwent radiotherapy but in the text there are not informations about it. If radiotherapy was performed it could be interesting to know reasons, volumes and doses.

Lastly, how the subsequent follow up was performed and with which imaging? 

Author Response

This is a very interesting case report describing a rare subtype of cervical cancer, that it is important to know.

The story of patients, diagnosis and treatment are well described except for radiotherapy: in the abstract it is reported that patient underwent radiotherapy but in the text there are not informations about it. If radiotherapy was performed it could be interesting to know reasons, volumes and doses.

Lastly, how the subsequent follow up was performed and with which imaging?

Authors’reply: Thank you very much for reviewing this article. I greatly appreciate you taking the time to review it!

As gynaecologists we thought that the patient will have chemoradiotherapy. That’s why we mention in the abstract that the patient was referred for subsequent chemotherapy and radiation therapy. The tumor board decided to perform only chemotherapy because of the metastasis in the lymph nodes. So, we fixed it in the abstract.

We performed the whole body CT after the fourth course of chemotherapy and we notice the tumor reduction. The case report is written after this CT